# Smoking Habits among College Students at a Public University in Riyadh, Saudi Arabia

**DOI:** 10.3390/ijerph191811557

**Published:** 2022-09-14

**Authors:** Khalid A. Bin Abdulrahman, Hatem Ali Alghamdi, Rayan Sulaiman Alfaleh, Waleed Saleh Albishri, Walid Bandar Almuslamani, Abdulelah Murdhi Alshakrah, Hamad Mohammed Alsuwailem, Sultan Ali Alkhelaiwi

**Affiliations:** 1Department of Medical Education, College of Medicine, Imam Mohammad Ibn Saud Islamic University (IMSIU), Riyadh 13317-4233, Saudi Arabia; 2Department of Public Health, College of Medicine, Imam Mohammad Ibn Saud Islamic University (IMSIU), Riyadh 13317-4233, Saudi Arabia

**Keywords:** smoking, students, smoking habits, shisha, electronic cigarettes

## Abstract

Tobacco smoking is one of the leading risk factors for ill health and death worldwide. Adolescence is the starting age of smoking for most current smokers worldwide. This study aimed to explore the prevalence of tobacco, the habits of different types of former smokers, and their relationship to other specialties and sociodemographic data. **Methods:** This cross-sectional study was conducted at Imam Mohammad Ibn Saud Islamic University (IMSIU), Saudi Arabia. An online questionnaire was sent to students’ emails to assess their smoking prevalence and tobacco and nicotine product habits. **Results**: Of the 895 students in IMSIU who participated in our survey, most reported having never used/tried tobacco, representing (76.4%). Most of the students who smoke began to smoke within the last five years (46.4%), which strongly indicates that they started to smoke when they entered the university. When students were asked about the time they like to smoke, most reported that they smoke when they feel stressed/under pressure (57.1%). There was a strong relationship between having a family member who smokes and being a smoker (53.1%). **Conclusions**: The prevalence of cigarettes, electronic cigarettes, and Shisha was 18.3%, 5%, and 11%, respectively. Anti-smoking regulations at the university level should be periodically reviewed to ensure the effectiveness and efficiency of tobacco control strategies.

## 1. Introduction

Tobacco smoking is one of the main risk factors for ill health and death worldwide [1]. If tobacco use continues as a global health burden, 8 million people will die annually by 2030 [2]. It is a common habit among people of different ages and sexes worldwide. Furthermore, it was found to be the beginning year of smoking for most smokers around the world. In a similar study, the duration of smoking was inversely associated with the age of initiation [1].

This habit is considered a pandemic since it causes the deaths of 8 million people per year (WHO) [2].

Tobacco Atlas stated that more than 20,000 children (10–14 years of age) and 3.35 million adults (15+ years of age) continue to use tobacco every day in Saudi Arabia, and more than 7000 deaths occur every year due to tobacco-induced diseases in the [1] region.

The Saudi Arabian Global Youth Tobacco Survey of Students aged 13–15 reported that 9.5% of students were current Shisha smokers [3]. A recent cross-sectional study has reported that the prevalence of those who have tried electronic cigarettes at least once in their lives is around 26% in Saudi Arabia [4]. Furthermore, a recent meta-analysis showed that smoking among college students in the KSA was 17%. Saudi male students had a prevalence rate of 26%, while for Saudi female students, the prevalence was 5% [5]. Furthermore, in six or more studies, family smoking status and friend smoking status were the two main social factors that showed a statistical relationship with smoking behavior. Environmental factors were the least tested for an association with tobacco smoking and were indicated in only two studies [6,7,8]. It should be mentioned that the most crucial factors in avoiding smoking and quitting smoking in Saudi Arabia were religious and health considerations [9]. 

According to the Saudi Revised Anti-Smoking Law Royal Decree No (M/56), dated 17 May 2015 [10], the cultivation or manufacturing of tobacco and its derivatives are prohibited in Saudi Arabia. Smoking shall mean the use of tobacco and its derivatives, such as cigarettes, cigars, tobacco leaves, tobacco molasses, and any product containing tobacco, whether in the form of cigarettes or cigars, by using a pipe or shisha, or by sniffing or chewing, or any other method. Smoking shall be prohibited at the following places: 1. Areas and yards surrounding mosques; 2. Ministries, government agencies, public institutions, their branches, and other public entities. 3. Educational, health, sport, cultural, and social institutions, as well as charities; 4. Work areas in companies, institutions, organizations, factories, banks, and the like; 5. Public means of transportation (land, air, or sea), as prescribed in the implementation regulations; 6. Places to produce, process, and packaging food, foodstuffs, and beverages; 7. Sites for producing, transporting, distributing, and refining petroleum, its derivatives, and fuel and gas stations; 8. Warehouses, elevators, and lavatories; and 9. Public places are not mentioned in the preceding paragraphs. If these places allocate smoking areas, these areas shall be isolated and restricted and not accessible to people under 18 years of age [10]. 

Although this topic has been addressed worldwide, studies on smoking among university students in Saudi Arabia need further exploration. The study aimed to explore the prevalence of tobacco, the habits of using different types of tobacco and nicotine products, and their relationship with other specialties and sociodemographic data. 

## 2. Materials and Methods 

### 2.1. Study Design

A cross-sectional observational study targeted college students from Imam Mohammad Ibn Saud Islamic University, including the humanities, sciences, and medicine. 

### 2.2. Participants and Sampling

All 6000+ students at Imam Mohammad Ibn Saud Islamic University from the streams of Humanities, Science, and Medicine were invited by email through the deanship of information technology. Two reminder emails and SMS weblink messages were sent to enhance the response rate. Data was collected from the students who responded from the colleges targeted. The sample size was calculated using Raosoft (Raosoft Inc., Seattle, WA, USA) based on a confidence level of 95% and a 5% margin of error. The total sample size calculated was 772. Additionally, 1200 students out of 6000 (20%) were randomly invited using convenient sampling to participate in the study (700 out of 3500 from humanitarian science, 300 out of 1500 from the science stream, and 200 out of 1000 from medicine).

### 2.3. Study Measurements

The questionnaire was designed to study university students’ exclusive cigarette smoking habits, tobacco used/tried, smoking behavior, different types of tobacco and nicotine products, and their relation to other variables and sociodemographic data.

The questionnaire consists of two parts. Part I addressed sociodemographic data, including age, sex, nationality, university name, and education year. Then, yes/no questions about smoking, electronic cigarettes, Shisha, and any first relative who smokes.

Part II covered smoking habits and behaviors, duration of smoking per year, number of cigarettes per day, shisha smoking per day, current smoking situation, preferred smoking times, the amount of money spent on tobacco per week, the level of enjoyment while smoking, the age when they started smoking, any intentions of quitting, for how long, and the ways of quitting. Students perceived the effectiveness of various measures to limit or prohibit smoking. 

### 2.4. Statistical Data Analysis

Means and standard deviations were used to describe continuous variables, and frequency and percentages were used to describe categorical variables. The Cronbach’s alpha reliability test showed that the 8-item questionnaire measuring the perceived effectiveness of various university students’ measures in reducing community smoking was reliable; Cronbach’s alpha = 0.80.

Multiple response dichotomy analysis was used to describe the questions measured with different options (tick all the questions that apply). The relative importance index (RII) was calculated for each indicator, which measures the perceived effectiveness of measures to prevent smoking using the formula below [11]. The items were ranked in ascending order of their R.I.I. (relative effectiveness) magnitude out of a hundred percent. The bivariate chi-square association test of association (χ^2^) was used to assess the correlation between the academic and professional characteristics of university students, as well as their smoking habits, with their current smoking behavior, and the independent samples *t*-test was used to assess the statistical significance of the mean difference in the metric variables between the levels of binary variables. Multivariate logistic regression analysis was used to determine the associations between the demographic and academic characteristics and the odds of being current smokers. The association of predictors with smoking behavior among students was expressed as an odds ratio (OR) with a 95% confidence interval. The SPSS IBM V21 program (IBM Corporation, Somers, NY, USA) and excel program were used for data analysis, and the alpha significance level was considered 0.050.

## 3. Results

Of the 1200 students invited to participate in the study, 895 (74.5%) completed the online survey. About two-thirds (65.8%) were male students. Most of the participants were Saudi nationals Table 1. 

Table 2 shows the smoking habits and the types of smoke used. The prevalence of smoking was 18.3%. At the same time, only 5% were current users of e-cigarettes. In addition, 11.8% were shisha smokers. Furthermore, 53.1% indicated that one or more of their family members were smokers.

The smoker and former smoker behavior of university students are shown in Table 3. To begin with, the analytical findings on the duration of smoking showed that about half (48.4%) smoked for 1–5 years. Furthermore, when asked to indicate how many cigarettes they had smoked on average, 49.3% smoked between 11 and 20 cigarettes. In addition, medical students were asked to describe their current smoking situation regarding the regularity of smoking: the analysis showed that 22.3% were former smokers. In comparison, 21.3% identified themselves as current and regular smokers. 

Stress and pressure appear to be the preferable time to smoke among university students (57.1%). 

The perceived reasons for becoming a smoker are presented in Figure 1. Smoking (48%), controlling anxiety (37.7%), and friends (34%) were the most common reasons for becoming a smoker.

The methods and motivations for quitting smoking are shown in Figure 2. The most common strategies to quit smoking were instantaneous and nicotine alternatives (59%).

Generally, according to the multivariate logistic regression model (Table 4), the analysis showed that male university students were predicted to be 7.33 times more likely to be smokers/former smokers than women, *p* < 0.001. The analysis model was statistically significant overall, and its predicted probability of student smoking behavior predicted the students smoking behavior well, as evidenced by a model area under the ROC equal to 86.7%, which is a good value.

Taking into account the other predictors in the analysis as controlled, male students are predicted to be significantly more likely to be smokers than females in all colleges (Figure 3). Furthermore, the analysis model suggested that the age converged slightly positively with their chances of being smokers, OR = 1.33 times higher; however, the association was not statistically significant at 0.136, accounting for the other predictors in the analysis. Furthermore, the multivariate analysis model showed that university students from various colleges differed significantly in their odds of being smokers, *p* = 0.006. However, the analysis showed that compared to medical students, humanities students were predicted to be 2.25 times more likely to be smokers/former smokers on average, *p* = 0.006. Furthermore, science students were predicted to be 1.36 times more likely to be smokers/former smokers than medical students, *p* = 0.003 (Figure 3). 

For example, the expected probability of smoking among medical students is less than that of students from colleges of sciences and humanities for both genders, accounting for the other predictors in the analysis. However, the year did not correlate significantly with current/former smoking behavior, *p* = 0.408. The analysis model revealed that the students did not differ significantly from those who had never used/tried e-cigarettes regarding their chances of being smokers, *p* = 0.166. However, they appear slightly more likely (OR = 2.53, 95% CI: 0.681:9.36), but the difference was not statistically significant. However, current smokers were expected to be significantly more likely present/former smokers (OR = 2.84 times higher) than those who had never used e-cigarettes (*p* = 0.008).

Furthermore, university students’ use of shisha was significantly correlated with their current and/or former smoking behaviors. Former shisha users were predicted to be 16.89 times more likely current cigarette smokers than those who never used/tried shisha, *p* < 0.001. Furthermore, students currently using shisha were predicted to be 6.67 times significantly more likely to be former/current cigarette smokers than those who had never used/tried shisha on average, *p* < 0.001. However, the analysis model showed that having a family member who is also a smoker correlated significantly with the student’s odds of being a former smoker / current smoker. Furthermore, it was predicted that students with one or more family members who are smokers were significantly more likely (2.63 times higher) to be smokers/former smokers than those without family members, on average, *p* < 0.001. However, students who stated that they were unsure about the smoking behavior of their family members did not differ significantly in their odds of being current/former cigarette smokers compared to those who did not have smoking family members as such, *p* = 0.0153, accounting for the other predictors in the analysis model. 

## 4. Discussion

The prevalence of current tobacco smokers among university students is 18.3%. Only 5% of college students were e-cigarette users. This could be attributed to the fact that e-cigarettes are more expensive, and the cost usually is not affordable for college students [12,13].

The prevalence of current shisha smokers (Hookah) was 11.8%. Unfortunately, some students had misconceptions about shisha. They thought it did not have nicotine, as it has a nice flavor and smells good; therefore, it is not as harmful as cigarettes [14,15,16]. 

Moreover, recent studies in some Arab countries gave different smoking rates among students, which is alarming, such as in KSA, among applied medical science college students (72%) [12,17]. This result could be explained on the basis that our university is an Islamic university, and therefore, the students are more likely to follow the Islamic instructions regarding smoking. 

Students with one or more family members who are smokers were predicted to be significantly more likely (2.63 times higher) to be cigarette smokers/former smokers than those without family members on average, *p* < 0.001. The result is logical, as children are affected by their role models, such as their parents, older brother, or sister [13]. Therefore, they are more likely to copy this habit and become smokers [3,6,15]. 

The duration of smoking showed that most of them smoked for 1 to 5 years. This result is reasonable, as this habit is usually acquired in high school and college settings. 

Regarding the number of cigarettes smoked daily, 49.3% of the students smoked between 11 and 20. This is considered high because they almost finish a whole pack each day. This could be attributed to college students being stressed by their studies and family and social obligations. Therefore, they think it helps relieve tension [15,16].

It is predicted that male university students are 7.33 times more likely to be smokers/former smokers than females, *p* < 0.001. This could be justified by the fact that male smokers tend to encourage their friends to smoke so that they can have things in common. Therefore, they spend much time together smoking in coffee shops and other public places. However, women tend to be more cautious and concerned about their health [15,18].

Compared to medical students, humanities students were predicted to be 2.25 times more likely to be smokers/former smokers, on average, *p* = 0.006. Furthermore, college science students were expected to be 1.36 times more likely to be smokers/former smokers than medical students, *p* = 0.003. This could be explained by the fact that students in the college of health education understand the risk factors for smoking. Therefore, medical students were the least in that group [19].

Former shisha users were 16.89 times more likely to be former smokers/current cigarette smokers than those who had never used/tried used shisha, *p* < 0.001. You are expected to find someone who smokes shisha or cigarettes, as it is a habit and an addiction. Once you have attached yourself to one, it is easy to go for the other. 

Despite the strict laws adopted by the government at all levels, including higher education and public institutions, to control tobacco and reduce the rates of its various uses, smoking habits are still increasing. It may be noted that awareness campaigns targeting the age groups that use the most tobacco products, such as high school students and early university students, have positive effects if reinforced by the availability of smoking cessation clinics within the corridors of educational institutions [20,21,22].

Recall bias is one of the significant limitations of such cross-sectional studies. Furthermore, the study participants were college students from one public university, which may limit its generalizability.

## 5. Conclusions

The prevalence of cigarettes, electronic cigarettes, and shisha was 18.3%, 5%, and 11%, respectively. Anti-smoking regulations at the university level should be periodically reviewed to ensure the effectiveness and efficiency of tobacco control strategies.

## Figures and Tables

**Figure 1 ijerph-19-11557-f001:**
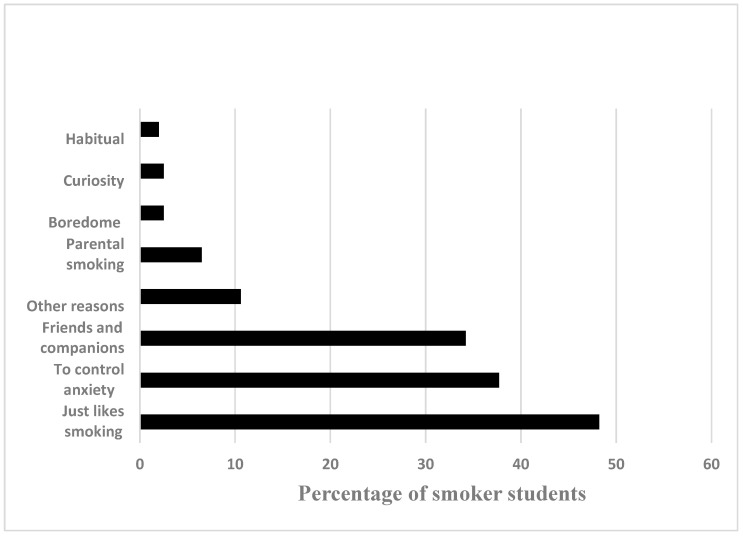
The perceived reasons for becoming a smoker.

**Figure 2 ijerph-19-11557-f002:**
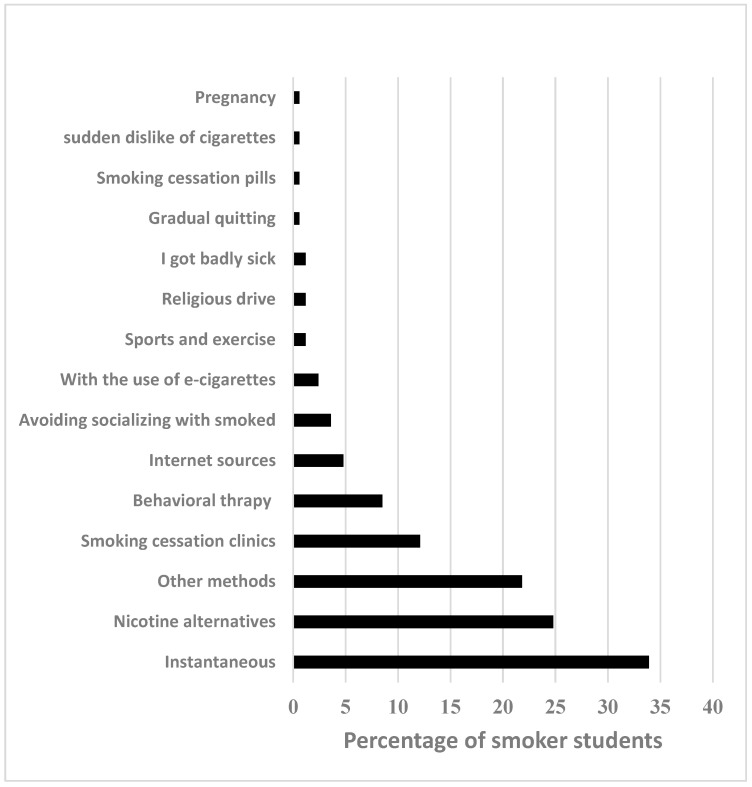
Methods/motivations for quitting cigarettes.

**Figure 3 ijerph-19-11557-f003:**
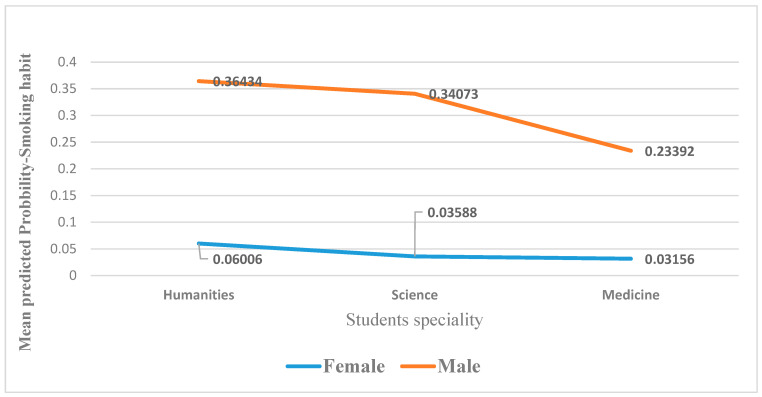
The association between student specialty and sex with their mean predicted probability of a positive history of smoking habits.

**Table 1 ijerph-19-11557-t001:** Descriptive characteristics of university students. N = 895.

	Frequency	Percentage (%)
**Sex**		
Female	306	34.2
Male	589	65.8
**Age group—years**		
18–22 years	562	62.8
23–26 years	272	30.4
>26	61	6.8
**Nationality**		
Non-Saudi	33	3.7
Saudi	862	96.3
**College**		
Humanities science	586	63.5
Science streams college (engineering + computer)	153	17.1
Medicine	174	19.4
**Education year**		
Junior (entry to 3rd year)	368	41.1
Intermediate (4th–5th year)	256	28.6
Senior (6th year or higher)	271	30.3

**Table 2 ijerph-19-11557-t002:** Current smoking habits.

	Frequency	Percentage (%)
**In general, are you a smoker? Or do you smoke any kind of tobacco?**		
Never used/tried	684	76.4
Former smoker	47	5.3
Current smoker	164	18.3
**Do you use e-Cigarettes?**		
Never used/tried	835	93.3
Former user	15	1.7
Currently smoking e-Cigarettes	45	5
Do you use Shisha (hookah)		
Never used/tried	745	83.2
Former Shisha smoker	44	4.9
Current shisha Smoker	106	11.8
**Do you have a family member who is a smoker?**		
No	389	43.5
Unsure	31	3.5
Yes	475	53.1

**Table 3 ijerph-19-11557-t003:** Descriptive statistics of university students’ current and former smoking behavior, N = 211.

	Frequency	Percentage (%)
**For how many years did you smoke?**		
1–5 years	98	48.4
6–10 years	60	26.4
11–16 years	20	9.5
>16	33	15.6
**How many cigarettes do you smoke per day?**		
Not sure/Not answered	6	2.8
≤ten cigarettes per day	68	32.2
11–20 Cigarettes per day	104	49.3
21–30 Cigarettes per day	22	10.4
>30 cigarettes per day	11	5.2
**Describe your current smoking situation**		
Former smoker	47	22.3
Smoked at one time or another	28	13.3
I smoke Irregularly	40	19
I am trying to quit cigarettes	51	24.2
I currently smoke regularly	45	21.3
**When do you like to smoke/What are the times you want to smoke, n = 203**		
I smoke most of the day times	112	55.2
Other times	11	5.4
I like to smoke after meals/coffee	97	47.8
I smoke when I feel stressed/under pressure	116	57.1
I smoke when I meet friends who smoke too	106	52.2
**Did you try to quit smoking?**		
I am a former smoker	47	22.3
No	45	21.3
Yes	119	56.4
**For how long did your quit period last**		
Not sure/Not answered	58	27.5
A week or less	41	19.4
From more than a week to less than a month	57	27
1–3 Months	33	15.6
4–6 Months	8	3.8
7–9 Months	2	0.9
10–12 Months	3	1.4
>12 Months	9	4.3
**How much Saudi riyal did you spend on buying cigarettes per week?**		
Less than 100 Saudi riyals	56	26.5
100–200 Saudi riyals	117	55.5
300–500 Saudi riyals	25	11.8
Greater than 500 Saudi riyals	13	6.2
**Rate your enjoyment level of cigarettes smoking**		
I don’t enjoy when smoking at all	31	14.7
I feel a slight joy when I smoke cigarettes	118	55.9
I have a lot of fun when I smoke cigarettes	62	29.4
**How old were you when you had your first cigarettes?**		
Can’t remember	7	3.3
Less than 11 years old	18	8.5
11–15 years old	68	32.2
16–19 years old	87	41.2
20 years old or older	31	14.7
**Whenever you were sick, did you still smoke?**		
No	100	47.4
Sometimes	78	37
Yes	33	15.6
**While smoking, did you realize that smoking is harmful to your health?**		
No	11	5.2
Unsure/Do Not know	8	3.8
Yes	192	91

**Table 4 ijerph-19-11557-t004:** Multivariate logistic binary regression analysis of university students’ smoking behaviors. N = 895.

	Adjusted Odds Ratio	95% CI for OR		*p*-Value
		Lower	Upper	
Students’ Sex = Male	7.325	3.982	13.476	<0.001
Student’s age	1.329	0.915	1.929	0.135
Students’ nationality	2.468	0.696	8.744	0.162
Stream = Medicine				0.006
Stream = Humanities	2.252	1.325	3.830	0.003
Stream = Science	1.360	0.657	2.817	0.408
Students’ education year	1.113	0.847	1.462	0.441
E-cigarette use = Never used/tried				0.014
E-cigarette use = Former	2.526	0.681	9.361	0.166
E-cigarette use = Current	2.842	1.314	6.146	0.008
Shisha use = Never used/tried				<0.001
Shisha use = Former	16.892	6.886	41.440	<0.001
Shisha use = Current	6.673	3.993	11.150	<0.001
Smoker family members = No				<0.001
Smoker family members = unsure	2.175	0.749	6.316	0.153
Smoker family members = Yes	2.628	1.729	3.994	<0.001
Constant	0.002			<0.001

## Data Availability

Data supporting this study’s findings are available from the corresponding author, K.A.B., upon reasonable request.

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
