# Peer review of "Smoking Habits among College Students at a Public University in Riyadh, Saudi Arabia"

_ijerph, 2022, doi:10.3390/ijerph191811557_

Round 1

Reviewer 1 Report

General comments

This manuscript aimed to explore the prevalence and correlates of cigarette smoking, hookah and e-cigarette use among Islamic university students in Riyadh, Saudi Arabia. Although the topic of this manuscript would be interesting and could have an important added value to the relevant national and regional scientific literature, in overall, the manuscript is poorly written.

The content of the manuscript does not meet with the STROBE criteria for observational (including cross-sectional) studies. In particular, Methods section is poorly written. Detailed and clear information about the survey procedure and participants is missing. Detailed description of measurements applied and involved in the current analytical sample is also missing. Dependent, covariate, and confounder variables were not presented clearly.

Citations in the text and Reference list editing are incorrect. Please follow the reference requirements of IJERPH Instruction for Authors.

Please unify the terms for tobacco product and smoking status. Please consider changing “smoking material” to “tobacco and nicotine products”, “passed” or “past smokers” to “former smokers”.

Thorough English language review and text editing would be needed before submitting a manuscript.

I encourage the Authors revising their manuscript according to my above and my further specific comments.

Specific Comments

Abstract

Please revise it after correcting the manuscript.

Introduction

One of the major weakness of this manuscript is the Introduction which is short and lacks important scientific background of the topic.

Please include national data about the prevalence of tobacco and nicotine product use (cigarette smoking, hookah use, e-cigarette use) among adolescents and adults.

Moreover, summarize shortly existing findings about tobacco and nicotine product use among college/university students in Saudi Arabia. Include traditional/religious factors that may influence tobacco use behaviors and may cause differences in it among youth.

Identify and review thoroughly relevant national studies. Several English language local studies were not included, so update your literature search.

Finally, highlight that why your present study is needed, what are your hypothesis, and according to these, clearly state your specific objectives.

Methods

Please follow the STROBE statements and checklist when writing Methods section.

In subsection 2.2., describe original sample size, sample size of respondents as well as response rate, and the final analytical sample size. Were there any eligibility criteria, e.g. age?

Please consider changing the title of subsection 2.3. “Study questionnaire” to “Measurements”. In this subsection, clearly define the dependent variable, that is, who were considered to be current or former smokers. Also define what you mean “smokers” (exclusive cigarette smokers, tobacco users?).

Lines 92-93: “student's perceived effectiveness of various measures” were not described in subsection 2.3. Please explain it in 2.3.

Clearly describe statistical analyses, especially the logistic regression model. Currently, it is unclear whether the Authors applied both unadjusted and adjusted models, what were the dependent, control, and covariate variables.

The Authors should keep in mind that the term „multivariate regression” is used for the analysis assuming multiply dependent variables. In case when more than one independent variable is used in the model, the adjective ‘multivariable’ or ‘multiple’ is adequate (please see Hidalgo, B., & Goodman, M. (2013). Multivariate or multivariable regression? AJPH doi: 10.2105/AJPH.2012.300897)

Results

Please change “descriptive analysis” to “descriptive characteristics…(of the sample)” in this section.

Lines 130-134. Here the Authors state that medical students were asked about their current smoking frequency. Why just medical students were asked? In Table 1, there were 174 medical students in the sample, why according to Table 3, n=211 medical students who were all current/former smokers? It is confusing so please revise.

Lines 140-141: Please revise this part of the sentence.

Figure 5 was mentioned in the text, although there is not any figure in the manuscript. Please include figure if exist.

Lines 157-160. This sentence is almost the same like the previous sentence. Revise it or omit.

Line 166-169. This sentence is confusing, please revise it.

In Table 1, please consider presenting your results by sex and also display p-values of statistical analyses. That is, columns: Total, Males, n (%); Females, n (%); p-value. Revise the title of Table 1. At “Level of study” variable, it is unclear what the meaning of "level" is? Does it refers to semesters or educational years? Please explain at Methods.

Table 2. Tobacco/nicotine use habits by sex is missing from Table 2. However, it would be valuable to present. Please revise this table and include such results. The variable text “In general, Are you a smoker? Or Do you smoke any kind of tobacco?” should be moved to measures and here present only „Smoking status”. Furthermore, it is unclear whether exclusive e-cigarette/shisha use or dual/ poly-use were presented. Please include a clear description about how tobacco/e-cigarette use status was assessed in the questionnaire. Revise the table and clearly indicate exclusive/dual use.

Table 3. In the title of this table, please indicate whether it is cigarette smoking or any other tobacco product use. It would be valuable to present results by current/former smoking status or provide an explanation why you decided to collapse current and former smoker categories.

Table 4. Students’ nationality was not mentioned in detailed in Methods or at descriptive results. Please include it. For respondents’ educational faculties, please use a subheading “educational faculty” instead of “stream”. Please indicate reference categories by “Ref.” in the table.

Revise logistic regression analysis (unadjusted, adjusted?) and also the text accordingly.

Discussion

Consider STROBE statements checklist when writing the Discussion. The first paragraph of the discussion should be the summary of key results with reference to study objectives.

Discussing study limitations is completely missing.

Discussing your results in light of results from similar studies and the generalizability of your findings would be needed. Please do not repeat study results in the Discussion.

The proportion of never smokers and never e-cigarette users is high compared to other studies among university students in Saudi Arabia. Please discuss these differences.

I am wondering whether the Authors have asked ever trial of different tobacco products and e-cigarettes. This is not clear according to the Measurement section. Never use also mean never trial of these products, separately, or never use means something different in this study?

Conclusion

Clear conclusions are missing. There is not any new scientific and practical added value in the Conclusion. It would be valuable including information about national tobacco control policies, especially the national situation on preventing tobacco and nicotine use among youth and future recommendation to curb the nicotine and tobacco epidemic among youth and young adulst.

Line 270. IRB Statement is missing.

Author Response

Response to reviewer-1

Open Review

English language and style

( ) Extensive editing of the English language and style required
( ) Moderate English changes required
(x) English language and style are fine/minor spell check required
( ) I don't feel qualified to judge the English language and style

Yes

Can be improved

Must be improved

Not applicable

Does the introduction provide sufficient background and include all relevant references?

( )

( )

(x)

( )

Are all the cited references relevant to the research?

( )

(x)

( )

( )

Is the research design appropriate?

( )

(x)

( )

( )

Are the methods adequately described?

( )

( )

(x)

( )

Are the results presented?

( )

( )

(x)

( )

Are the conclusions supported by the results?

( )

( )

(x)

( )

Comments and Suggestions for Authors

*General comments

This manuscript aimed to explore the prevalence and correlates of cigarette smoking, hookah, and e-cigarette use among Islamic university students in Riyadh, Saudi Arabia. Although the topic of this manuscript would be interesting and could have an important added value to the relevant national and regional scientific literature,  overall, the manuscript is poorly written.

The manuscript's content does not meet the STROBE criteria for observational (including cross-sectional) studies. In particular, the Methods section is poorly written. Detailed and clear information about the survey procedure and participants is missing. A detailed description of measurements applied and involved in the current analytical sample is also missing. Dependent, covariate, and confounder variables were not presented clearly.

Author Response:

Thank you very much for your comment. For updated information on the introduction, methods, results, and discussion sections. The manuscript's content matched the STROBE statement checklist of observational cross-sectional studies.

*Citations in the text and Reference list editing are incorrect. Please follow the reference requirements of the IJERPH Instruction for Authors.

Author Response:

Thank you very much for your comment. The citations in the text and reference list were updated to follow the reference requirements of the IJERPH Instruction for Authors.

*Please unify the terms for tobacco products and smoking status. Please consider changing “smoking material” to “tobacco and nicotine products”, “passed” or “past smokers” to “former smokers”.

Author Response:

Thank you very much for your comment. The term ‘smoking material’ has been changed to “tobacco and nicotine products throughout the text and tables.

*Thorough English language review and text editing would be needed before submitting a manuscript.

Author Response:

Thank you very much for your comment. The English editing has been double-checked. Moreover, the Scholars Editing Services carried out the English proofreading. Please, see the attached certificate of English Editing.

*Specific Comments

*Abstract

Please revise it after correcting the manuscript.

Author Response:

Thank you very much for your comment. The abstract has been revised.

*Introduction

One of the major weaknesses of this manuscript is the Introduction which is short and lacks important scientific background on the topic.

Please include national data about the prevalence of tobacco and nicotine product use (cigarette smoking, hookah use, e-cigarette use) among adolescents and adults.

Moreover, summarize shortly existing findings about tobacco and nicotine product use among college/university students in Saudi Arabia. Include traditional/religious factors that may influence tobacco use behaviors and may cause differences in it among youth. Identify and review thoroughly relevant national studies. Several English language local studies were not included, so update your literature search.

Author Response:

Thank you very much for your comment. Two new paragraphs were added in the introduction in lines 47-57 about tobacco use among Saudi college students and health/religious factors that may influence tobacco use. Five new citations (4-9) were added to the reference list.

*Finally, highlight why your present study is needed, what are your hypothesis, and according to these, clearly state your specific objectives.

Author Response:

Thank you very much for your comment. The last paragraph of the introduction was revised to include the rationale and specific objective of the study. Please, see lines 76-79.

*Methods

Follow the STROBE statements and checklist when writing the Methods section.

Author Response:

Thank you very much for your comment. The methods section was rephrased according to the STROBE statement checklist.

*In subsection 2.2., describe the original sample size, the sample size of respondents as well as response rate, and the final analytical sample size. Were there any eligibility criteria, e.g. age?

Author Response:

Thank you very much for your comment. One thousand two hundred (1200) students were randomly invited to participate in the study ( 700 from humanitarian science, 300 from the science stream, and 200 from medicine). Eight hundred and ninety-five (895) students responded to the online survey, that is, the response rate was (74.5%). Please, see lines 91-93 in the methods section and 137-138 in the results section.

*Please consider changing the title of subsection 2.3. “Study questionnaire” to “Measurements”. In this subsection, clearly define the dependent variable, that is, who were considered to be current or former smokers. Also define what you mean by “smokers” (exclusive cigarette smokers, tobacco users?).

Lines 92-93: “student's perceived effectiveness of various measures” were not described in subsection 2.3. Please explain it in 2.3.

Author Response:

Thank you very much for your comment. Done; the perceived effectiveness of each measure will be presented in another paper. please see lines 95 and 107-109.

8 Clearly describe statistical analyses, especially the logistic regression model. Currently, it is unclear whether the Authors applied both unadjusted and adjusted models, and what were the dependent, control, and covariate variables.

Author Response:

Thank you very much for your comment. The statistical data analysis section describes all the statistical tests; please see section 2.4 lines 111-131. Multivariate logistic regression analysis was performed to estimate the multivariate-adjusted odds ratios and their associated confidence interval for students ( all students) odds of being a current smoker. The interpretation of the findings for the multivariate analysis model shows the comparison reference groups for each included variable, and this was clarified in the interpretation. The bivariate analysis with independent samples t-test and the chi-squared test of independence were used as a background descriptive introductory analysis, but we did not publish them in the paper,  and we used bivariate analysis findings to select the most significant and relevant factors to include in the multivariate analysis.

* The authors should keep in mind that the term ‘multivariate regression is used for the analysis assuming multiple dependent variables. In case more than one independent variable is used in the model, the adjective ‘multivariable’ or ‘multiple’ is adequate (see Hidalgo, B., & Goodman, M. (2013). Multivariate or multivariable regression? AJPH doi: 10.2105/AJPH.2012.300897)

Author Response:

Thank you very much for your comment. The word multivariable has been used to describe the title of the model. Please see lines 189, 191, 193, table 4 footnote, 202, 205, 233, 245, 253.

*Results

Change ‘descriptive analysis’ to “descriptive characteristics…(of the sample)’ in this section.

Author Response:

Thank you very much for your comment. The term ‘descriptive analysis has been replaced by “descriptive characteristics. See line 140 in the results section.

*Lines 130-134. Here, the authors state that medical students were asked about their current smoking frequency. Why were only medical students asked? In Table 1, there were 174 medical students in the sample, which is why according to Table 3, n=211 medical students were all current/former smokers. It is confusing, so please revise.

Author Response:

Thank you for noting that the word medical is “mistyped’; the analysis is carried out on all students in that section, not only medical students. The word medical has been deleted. Please, see Table 3 line 157.

*Lines 140-141: Please review this part of the sentence.

Author Response:

Thank you very much for your comment. The paragraph in the updated lines 183-185 has been modified as follows. “Generally, according to the multivariate logistic regression model (Table 4), the analysis showed that male university students were predicted to be 7.33 times more likely to be smokers/former smokers than females, p<0.001. When considering the other predictors in the analysis as controlled for, male students are predicted to be significantly more likely to be smokers than females across all colleges. ‘

*Figure 5 was mentioned in the text, although there is no figure in the manuscript. Please include a figure if exists.

Author Response:

Thank you very much for your comment. Figures 1-3 have been added in the results section. However, Figure 5 was stated by mistake.

*Lines 157-160. This sentence is almost the same as the previous sentence. The revision or omit.

Author Response:

Thank you very much for your comment. The sentence has been modified; please, see the updated lines 183-188. ‘Generally, according to the multivariate logistic regression model (Table 4), the analysis showed that male university students were predicted to be 7.33 times more likely to be smokers/former smokers than females, p<0.001. The analysis model was statistically significant overall, and its predicted probability of student smoking behavior predicted the students smoking behavior well, as evidenced by a model area under the R.O.C. equal to 86.7%, which is a good value.

* Lines 166-169. This sentence is confusing; please, revise it.

Author Response:

Thank you very much for your comment. The statement has been revised to be much clearer for international readers; please see lines 233-236 as follows;

“Moreover, the university students' use of Shisha by university students was significantly correlated with their current and/or former smokers. Former shisha users were predicted to be 16.89 times more likely to be current cigarette smokers than those who never used Shisha, p<0.001.”

*In Table 1, consider presenting your results by sex and also displaying the p-values of statistical analyses. That is, columns: Total, Males, n (%); Females, n (%); p-value. Revise the title of Table 1. In the variable level of study, it is unclear what the meaning of ‘level’ is. Does it refer to semesters or educational years? Explain at Methods.

Author Response:

Thank you very much for your comment. The level of study has been replaced by educational year in the methods section and Table 1; please see lines 225 and tables 1.

*Table 2. The habits of tobacco/nicotine use by sex are missing from Table 2. However, it would be valuable to present this paper. Please, review this table and include such results. The variable text ‘In general, Are you a smoker? Or Do you smoke any kind of tobacco? ‘ should be moved to measures, and here present only „Smoking status”. Furthermore, it is unclear whether exclusive use or dual/ poly-use were presented. Include a clear description of how tobacco / electronic cigarette use status was assessed in the questionnaire. Revise the table and indicate exclusive/dual use.

Author Response:

Thank you very much for your comment. The title of Table 2 is corrected to be; The University student's current smoking habits. Please, see line 146. However, the association between sex and smoking was considered in the bivariate and multivariate analyzes in table-4.

*Table 3. In the title of this table, indicate whether it is cigarette smoking or any other use of tobacco product use. It would be valuable to present results by current/former smoking status or provide an explanation of why you decided to collapse current and former smoker categories.

Author Response:

Thank you very much for your comment. The dependent variable in this analysis was the smoking behavior of cigarettes : ( current or former).  Please note the dependent variable was specified in the logistic regression analysis table 4 - footnote just at the bottom.

*Table 4. Students’ nationality was not mentioned in detail in the Methods or descriptive results. Please include it. For the educational faculties, please use the subheading ‘educational faculties’ instead of “stream”. Indicate the reference categories by “Ref.” in the table.

Author Response:

Thank you very much for your comment. The student's nationality was described in Table 1. We have a few non-Saudi students =33 included in the study, most of the participants (96.35) were Saudi national students.

*Revise the logistic regression analysis (unadjusted, adjusted?) and also the text accordingly.

Author Response:

Thank you very much for your comment. The word ( unadjusted)  was not used in the manuscript! But rather the multivariate-adjusted association was used. The analysis considers the multivariate-adjusted associations; in the analysis section, it is indicated that the multivariate analysis focused on assessing the combined & individual associations in the multivariate modeling analysis. 

Also, kindly note in the interpretation of the logistic regression analysis, we interpreted the findings in the correct way it should by stating ( after considering the other predictor variables as accounted for in the multivariable analysis ) that this interpretation intends to show that it is a multivariable/multivariate-adjusted odds ratio/ association.

Discussion

Consider the STROBE statements checklist when writing the Discussion. The first paragraph of a discussion should be the summary of key results with reference to the study objectives.

Author Response:

Thank you very much for your comment. The discussion has been revised according to the STROBE statement checklist.

Discussing the limitations of the study is completely missing.

Author Response:

Thank you very much for your comment. A study limitation was added at the end of the discussion section. See lines 282-284.

Discussing your results in light of results from similar studies and the generalizability of your findings would be necessary. Please do not repeat the study results in the Discussion.

Author Response:

Thank you very much for your comment. The study discussion has been revised according to the STROBE statement checklist.

The proportion of never-smokers and never-e-cigarette users is high compared to other studies among university students in Saudi Arabia. Please discuss these differences.

Author Response:

Thank you very much for your comment. Two paragraphs in the discussion section have addressed your issue. See lines 299-301.

I am wondering whether the authors have asked for ever trial of different tobacco products and e-cigarettes. This is not clear according to the Measurement section. Never use also means never trial of these products, separately, or never use means something different in this study?

Author Response:

Thank you very much for your comment. In the text (never use) means (never trial) as well. Please, see line 97 in the methods section and out through the text and tables 1 and 4.

Conclusion

Clear conclusions are missing. There is no new scientific and practical added value in the Conclusions. It would be valuable to include information on national tobacco control policies, especially the national situation on preventing tobacco and nicotine use among youth and future recommendations to curb the nicotine and tobacco epidemic among youth and young adults.

Author Response:

Thank you very much for your comment. The conclusion has been revised. Please, see lines 303-307.

Line 270. The IRB statement is missing.

Author Response:

Thank you very much for your comment. The ethical consideration statement has been added after the conclusion on lines 310-316.

Submission Date

24 July 2022

Date of this review

09 Aug 2022 14:54:20

End of the response to the editor and reviewer’s report

Reviewer 2 Report

The abstract should not be long. Ideally, it should not exceed 250 words.  Phrases: This habit is considered a pandemic as it causes the death of 8 million people each year 16 (WHO). And: It is expected that if tobacco use continues, 8 million people will die yearly by 2030.  These phrases should be reconciled.  Readers will note that no change is expected by 2030.

It is advisable to choose one of the editor's recommended ways of citing literature sources.

In addition to traditional smoking, it is desirable to mention in the "Introduction" the epidemiology of e-smoking in Saudi Arabia. E.g. Althobaiti NK, Mahfouz MEM. Prevalence of Electronic Cigarette Use in Saudi Arabia. Cureus. 2022 Jun 7;14(6):e25731.

Provide the Cronbach's Alpha score for the questions in the questionnaire. Was the use of The Relative Importance Index (RII) justified? After all, the results do not mention it.

It is advisable to avoid references to literature sources in the conclusion. After all, this is the conclusion suggested by the authors.

Author Response

Reviewer-2

Open Review

English Language and style

( ) Extensive editing of English language and style required( ) Moderate English changes required(x) English language and style are fine/minor spell check required( ) I do not feel qualified to judge the English language and style

Yes

Can be improved

Must be improved

Not applicable

Does the Introduction provide sufficient background and include all relevant references?

( )

( )

(x)

( )

Are all the references cited relevant to the research?

( )

(x)

( )

( )

Is the research design appropriate?

( )

(x)

( )

( )

Are the methods adequately described?

( )

( )

(x)

( )

Are the results presented?

( )

(x)

( )

( )

Are the conclusions supported by the results?

( )

(x)

( )

( )

Comments and suggestions for Authors

*The abstract should not be long. Ideally, it should not exceed 250 words. 

Author Response:

Thank you very much for your comment. The abstract has been reduced to 237 words.

*Phrases: This habit is considered a pandemic as it causes the deaths of 8 million people each year 16 (WHO). And: It is expected that if tobacco use continues, 8 million people will die yearly by 2030.  These phrases should be reconciled.  Readers will note that no change is expected by 2030.

Author Response:

Thank you very much for your comment. The statement has been modified. See lines 35-37.

*It is advisable to choose one of the editors’ recommended ways of citing literature sources.

Author Response:

Thank you very much for your comment. Citation and references were according to the journal guidelines.

*In addition to traditional smoking, it is desirable to mention in the "Introduction" the epidemiology of e-smoking in Saudi Arabia. For example, Althobaiti NK, Mahfouz MEM. Prevalence of electronic cigarette use in Saudi Arabia. Cureus. 2022 Jun 7;14(6):e25731.

Author Response:

Thank you very much for your comment. Two paragraphs have been added to the introduction about the situation of smoking in Saudi Arabia. See lines 47-57.

*Provide the Cronbach's Alpha score for the questions in the questionnaire. Was the use of the relative importance index (RII) justified? After all, the results do not mention it.

Author Response:

Thank you very much for your comment. Cronbach's Alpha score has been stated in lines 113-115, and

*It is advisable to sources from avoiding references to literature in the conclusion. After all, this is the conclusion suggested by the authors.

 Author Response:

Thank you very much for your comment. In conclusion, was removed. Please, see the conclusion section on lines 303-307.

Submission Date

24 July 2022

Date of this review

07 Aug. 2022 19:33:41

End of the response to the editor and reviewer’s report

Reviewer 3 Report

The study topic is important and up-to-date.

To improve the scientific soundness of this manuscript, the following changes should be considered:

1. The manuscript requires language revisions to improve the wording and style of the text.

2. The Abstract section should be revised - please provide more informative data and clarify which data refers to the whole groups and which to the current smokers etc.

3. The introduction section is too limited. Please provide a comprehensive overview of tobacco use and related issues in Saudi Arabia. 

4. Data were collected in 2019. The results are quite old, as during the previous 3 years a COVID-19 pandemic has had a significant impact on tobacco use behaviors.

5. Data on response rate and the total number of students at the university should be provided. 

6. Questions used to assess smoking status are unclear. Please justify why these questions were used and how the smoking status was assessed in previous studies. 

7. Please clearly define "n" in rows to clarify whether the percentage refers to the whole population or to a selected group of smokers. 

8. Please provide data on tobacco control law in Saudi Arabia as well as anti-tobacco actions targeted to achieve smoke-free universities.

9. This study has numerous limitations. Please provide the limitations section.

10. The sample size is a conventional sample. The results can not be generalized to the whole population of students in Saudi Arabia. Please adjust the concussions to your own findings and avoid overwhelming conclusions. 

Author Response

Reviewer-3

Open Review

English Language and style

( ) Extensive editing of English language and style required(x) Moderate English changes required( ) English language and style are fine/minor spell check required( ) I don't feel qualified to judge the English language and style

Yes

Can be improved

Must be improved

Not applicable

Does the Introduction provide sufficient background and include all relevant references?

( )

( )

(x)

( )

Are all the references cited relevant to the research?

( )

(x)

( )

( )

Is the research design appropriate?

( )

( )

(x)

( )

Are the methods adequately described?

( )

( )

(x)

( )

Are the results presented?

( )

( )

(x)

( )

Are the conclusions supported by the results?

( )

( )

(x)

( )

Comments and suggestions for Authors

*The study topic is important and up to date.

Author Response:

Thank you very much for your comment. 

To improve the scientific soundness of this manuscript, the following changes should be considered.

  1. The manuscript requires language revisions to improve the language and style of the text.

Author Response:

Thank you very much for your comment. The English editing has been double-checked. Moreover, the Scholars Editing Services carried out the English proofreading. Please, see the attached certificate of English Editing.

The Abstract section should be revised; to provide more informative data and clarify which data refers to the whole group and which to the current smokers etc.

Author Response:

Thank you very much for your comment. The abstract has been revised.

  1. The introduction section is too limited. Provide a comprehensive overview of tobacco use and related issues in Saudi Arabia. 

Author Response:

Thank you very much for your comment. Two paragraphs have been added to the introduction about the situation of smoking in Saudi Arabia. See lines 47-57.

  1. Data were collected in 2019. The results are quite old, as, during the previous 3 years, a COVID-19 pandemic has had a significant impact on tobacco use behaviors.

Author Response:

Thank you very much for your comment. We thought it was an advantage to conduct our survey before the pandemic. During the pandemic, most of the university courses were online.

  1. Data on the response rate and the total number of students at the university should be provided. 

Author Response:

Thank you very much for your comment. One thousand two hundred (1200) students were randomly invited to participate in the study ( 700 from humanitarian science, 300 from the science stream, and 200 from medicine). Eight hundred and ninety-five (895) students responded to the online survey, that is, the response rate was (74.5%). Please, see lines 91-93 in the methods section and 137-138 in the results section.

  1. The questions used to assess smoking status are unclear. Justify why these questions were used and how smoking status was assessed in previous studies. 

Author Response:

Thank you very much for your comment. The questions related to smoking status were clearly stated in the Methods section on lines 104-109.

  1. Please clearly define "n" in rows to clarify whether the percentage refers to the whole population or a selected group of smokers. 

Author Response:

Thank you very much for your comment.  Could you kindly state which table pertains to this comment because it is unclear? However, the multiple response dichotomies analysis was considered to describe the questions with more than an option, this analysis, in particular, does not produce a percentage that adds to a hundred percent as a ceiling, and that is acceptable ( if that was your concern for some percentages that does not add to 100%), however, I am not sure which tables do you kindly imply. 

  1. Provide data on tobacco control law in Saudi Arabia, as well as anti-tobacco actions targeted at achieving smoke-free universities.

              Author Response:

Thank you very much for your comment. The Saudi tobacco control law has been summarized in the Introduction section on lines 58-75.

  1. This study has numerous limitations. Please provide the limitation section.

Author Response:

Thank you very much for your comment. The limitation has been stated at the end of the discussion section on lines 299-301.

  1. The sample size is a conventional sample. The results cannot be generalized to the entire population of students in Saudi Arabia. Adjust concussions to your findings and avoid overwhelming conclusions. 

Author Response:

Thank you very much for your comment. The conclusion was rephrased according to your kind note. Please see lines 303-307.

Submission Date

24 July 2022

Date of this review

10 Aug. 2022 10:38:38

End of the response to the editor and reviewer’s report

Round 2

Reviewer 3 Report

The Authors applied some revisions, however, this study still requires major changes.

1. First of all, the language and style used in this study should be revised by the native English speaker. 

2. There is a lack of references in lines 57-73.

3. This is a cross-sectional survey on the prevalence of tobacco use among students. Response rate is crucial. How can we assess the scientific soundness of this manuscript without data on the percentage of students who were surveyed? E.G., the faculty has 5,000 students, and the Authors screened a convenience sample of 700 students. 

4. Study design, sampling methods, and distribution of the questionnaire should be presented. 

5. Conclusions are overwhelming and should be revised. Please provide informative conclusions based on your own findings.

Author Response

Reviewer 3 - Round 2 - ijerph-1856663

Open Review

English Language and style

( ) Extensive editing of the English language and style required
(x) Moderate English changes required
( ) English language and style are fine/minor spell check required
( ) I don't feel qualified to judge the English language and style

Yes

Can be improved

Must be improved

Not applicable

Does the introduction provide sufficient background and include all relevant references?

( )

(x)

( )

( )

Are all the cited references relevant to the research?

( )

( )

(x)

( )

Is the research design appropriate?

( )

(x)

( )

( )

Are the methods adequately described?

( )

(x)

( )

( )

Are the results clearly presented?

( )

(x)

( )

( )

Are the conclusions supported by the results?

( )

(x)

( )

( )

Comments and Suggestions for Authors

The Authors applied some revisions; however, this study still requires major changes.

  1. First of all, the language and style used in this study should be revised by the native English speaker. 

Author Response:

Thank you very much for your comment.  English editing has been double-checked. Moreover, the Scholars Editing Services carried out the English proofreading. Please, see the attached certificate of English Editing.

  1. There is a lack of references in lines 57-73.

Author Response:

Thank you very much for your comment. Reference 10 has been cited in lines 58 and 73. Moreover, it has been inserted into the reference list on line 350.

  1. This is a cross-sectional survey on the prevalence of tobacco use among students. The response rate is crucial. How can we assess the scientific soundness of this manuscript without data on the percentage of students who were surveyed? E.G., the faculty has 5,000 students, and the Authors screened a convenience sample of 700 students. 

Author Response:

Thank you very much for your comment. The total number of students in the three college streams has been inserted in lines 85 and lines 91 and 94.

  1. Study design, sampling methods, and questionnaire distribution should be presented. 

Author Response:

Thank you very much for your comment. The study design was presented in Section 2.1 in the method section on lines 81-83. The sampling methods and the distribution of the questionnaire were presented in Section 2.2 on lines 85-94.

  1. Conclusions are overwhelming and should be revised. Please provide informative conclusions based on your own findings.

Author Response:

Thank you very much for your comment. The conclusion has been revised. Please, see lines 299-301

Submission Date

24 July 2022

Date of this review

07 Sep 2022 08:36:07

End of the response to the editor and reviewer’s report
